# Bromelain Protects Critically Perfused Musculocutaneous Flap Tissue from Necrosis

**DOI:** 10.3390/biomedicines10061449

**Published:** 2022-06-19

**Authors:** Andrea Weinzierl, Yves Harder, Daniel Schmauss, Michael D. Menger, Matthias W. Laschke

**Affiliations:** 1Institute for Clinical & Experimental Surgery, Saarland University, 66421 Homburg, Germany; michael.menger@uks.eu (M.D.M.); matthias.laschke@uks.eu (M.W.L.); 2Department of Plastic, Reconstructive and Aesthetic Surgery, Ospedale Regionale di Lugano, Ente Ospedaliero Cantonale (EOC), 6900 Lugano, Switzerland; yves.harder@eoc.ch (Y.H.); schmauss.daniel@gmail.com (D.S.); 3Faculty of Biomedical Sciences, Università della Svizzera Italiana, 6900 Lugano, Switzerland

**Keywords:** bromelain, phytochemicals, random-pattern flap, necrosis, angiogenesis, nutrition, microcirculation, dorsal skinfold chamber, intravital fluorescence microscopy

## Abstract

Bromelain has previously been shown to prevent ischemia-induced necrosis in different types of tissues. In the present study, we, therefore, evaluated for the first time, the tissue-protective effects of bromelain in musculocutaneous flaps in mice. Adult C57BL/6N mice were randomly assigned to a bromelain treatment group and a control group. The animals were treated daily with intraperitoneal injections of 20 mg/kg bromelain or saline (control), starting 1 h before the flap elevation throughout a 10-day observation period. The random-pattern musculocutaneous flaps were raised on the backs of the animals and mounted into a dorsal skinfold chamber. Angiogenesis, nutritive blood perfusion and flap necrosis were quantitatively analyzed by means of repeated intravital fluorescence microscopy over 10 days after surgery. After the last microscopy, the flaps were harvested for additional histological and immunohistochemical analyses. Bromelain reduced necrosis of the critically perfused flap tissue by ~25%. The bromelain-treated flaps also exhibited a significantly higher functional microvessel density and an elevated formation of newly developed microvessels in the transition zone between the vital and necrotic tissues when compared to the controls. Immunohistochemical analyses demonstrated a markedly lower invasion of the myeloperoxidase-positive neutrophilic granulocytes and a significantly reduced number of cleaved caspase 3-positive apoptotic cells in the transition zone of bromelain-treated musculocutaneous flaps. These findings indicate that bromelain prevents flap necrosis by maintaining nutritive tissue perfusion and by suppressing ischemia-induced inflammation and apoptosis. Hence, bromelain may represent a promising compound to prevent ischemia-induced flap necrosis in clinical practice.

## 1. Introduction

The use of phytochemicals has a long-standing tradition in health care. Even nowadays, they play an important role in treating various ailments due to their high therapeutic efficiency and acceptability by patients with different pathologies. Up to 50% of the presently used drugs are based on molecules of natural origins, highlighting the fact that phytochemicals are still a valuable resource in modern medicine [1,2].

The enzyme complex bromelain is obtained from pineapple stems (*Ananas comosus*) and represents one example of a well-known phytochemical that was initially used as an over-the-counter anti-inflammatory agent but has now proven to be useful for various other indications [3,4,5]. In fact, bromelain has been applied in the therapy of hematomas, rheumatoid arthritis, thrombophlebitis, oral inflammation, diabetic ulcers, rectal and perirectal inflammation, angina pectoris, bronchitis, sinusitis, surgical traumas and pyelonephritis [6,7]. In all these cases, orally administered bromelain could reduce pain and swelling and facilitate faster healing [6,8]. The possible mechanisms mediating these beneficial effects are the dose-dependent modulation of bradykinin secretion [9], an improved microcirculatory flow due to the fibrinolytic and anticoagulant activity of the enzymes [10], the downregulation of inflammatory mediators [11] and the modulation of leukocyte–endothelial cell interactions [12]. Interestingly, bromelain has also been shown to protect hepatocytes from apoptosis after warm ischemia [13].

Due to the combination of anti-inflammatory, anti-apoptotic and anti-thrombotic properties, bromelain may be a valuable therapeutic compound to prevent ischemic damage in tissue flaps. Such flaps can consist of one or more tissue subtypes and are either raised on a vascular axis or as so-called random-pattern flaps. In the case of the random-pattern flaps, the blood perfusion is provided by the dermal plexus and/or musculocutaneous arterioles that pass through the base of the flaps [14]. The decreased arterial inflow places the tissue zones distant to the flap base at risk for ischemic complications, such as wound breakdown or necrosis. Depending on their thickness, tissue composition and regions of the body, random-pattern flaps are, therefore, planned according to certain length-to-width-ratios [15,16]. Furthermore, comorbidities like diabetes or lifestyle factors, such as active smoking habits, have to be considered in the planning of a flap. Nonetheless, despite careful preoperative planning, the occurrence rates of ischemic flap complications remain high. Pestana et al. [17] reported that in mastectomy flaps, more than 70% of the examined flaps demonstrated poor perfusion, as assessed by an intraoperative fluorescent angiography. Wound breakdown and flap necrosis, which results from such inadequate perfusion, will negatively impact patient morbidity and markedly increase treatment costs. Hence, promoting flap survival and preventing ischemia-induced tissue damage may considerably improve the clinical outcome of flap surgery and eventually benefit patient care in plastic surgery.

In the present study, we, therefore, wanted to analyze whether bromelain exerts beneficial tissue-protective effects on surgical flaps. For this purpose, we prepared random-pattern musculocutaneous flaps in the dorsal skinfold chamber of mice and administered daily 20 mg/kg bromelain or saline (control) by intraperitoneal injection starting 1 h before the flap elevation throughout a 10-day observation period. Vascularization, inflammation and flap necrosis were subsequently analyzed using intravital fluorescence microscopy, histology and immunohistochemistry.

## 2. Materials and Methods

### 2.1. Animals

The animal experiments were approved by the local governmental animal protection committee (permit number: 10/2020) and were conducted in accordance with the European legislation on the protection of animals (Directive 2010/63/EU) and the NIH Guidelines on the Care and Use of Laboratory Animals (NIH publication #85-23 Rev. 1985).

A total of 14 male C57BL/6N wild-type mice (Institute for Clinical and Experimental Surgery, Saarland University, Homburg, Germany) were used for the present study. The animals had an age of 12–24 weeks and a body weight of 26–30 g. During the experiments, the animals were kept in individual cages at a room temperature of 22–24 °C, with a relative humidity of 55–60% and a 12 h day–night cycle. Access to standard pellet chow (Altromin, Lage, Germany) and tap water was unrestricted.

### 2.2. Bromelain Treatment

Based on the study of Juhasz et al. [18], a daily intraperitoneal dose of 20 mg/kg bromelain (Bio Protect BV, Kerkrade, The Netherlands), dissolved in isotonic saline solution, was chosen for the present study because it has been shown to be effective and well-tolerated by animals. Seven mice were each randomly assigned to a bromelain group and a control group, which received daily intraperitoneal saline injections of an equal volume starting 1 h before the flap elevation throughout a 10-day observation period.

### 2.3. Anesthesia

The mice were placed under general anesthesia for the surgical elevation of the flap, the implantation of the dorsal skinfold chamber, and the subsequent intravital fluorescence microscopy. For this purpose, the animals received an intraperitoneal injection of ketamine (100 mg/kg of body weight; Ursotamin^®^; Serumwerke Bernburg, Bernburg, Germany) and xylazine (12 mg/kg of body weight; Rompun^®^; Bayer, Leverkusen, Germany). To prevent postoperative pain, all animals were additionally treated with subcutaneously injected buprenorphine hydrochloride (0.01 mg/kg of body weight; Temgesic^®^; RB Pharmaceuticals Limited, Slough, UK).

### 2.4. Dorsal Skinfold Chamber-Flap Model

A musculocutaneous flap was elevated on the back of each animal and mounted into a dorsal skinfold chamber (Irola Industriekomponenten GmbH & Co. KG, Schonach, Germany), as described previously in detail [19]. This model allows for repeated intravital fluorescence microscopy to study the microcirculation within the flap tissue (Figure 1A). In brief, after depilation of the dorsal skin, a musculocutaneous flap, measuring 15 mm (width) × 11 mm (length), was elevated perpendicular to the spine. The lateral wound margins were sutured back to the wound bed, and the dorsal skinfold, including the musculocutaneous flap, was fixed to one chamber frame. The second frame was covered with adhesive insulation foam to adequately seal the chamber and was then mounted to its counterpart. Accordingly, the flap was finally sandwiched between the two chamber frames, making it accessible for microscopic imaging through the observation window of the chamber. The window was subsequently closed with a cover glass that was fixed by means of a snap ring. Due to the chosen width-to-length-ratio, the distal portion of the flap was subject to acute persistent ischemia and the tissue developed roughly 50% necrosis without treatment (Figure 1B). After the preparation, the animals could recover for 24 h from anesthesia and surgery before the first microscopy. All animals tolerated the surgical interventions well, as evidenced by their normal food intake and behavior during the remaining observation period.

### 2.5. Intravital Fluorescence Microscopy

Intravital fluorescence microscopy was performed on days 1, 3, 5, 7 and 10 after the flap elevation. For this purpose, the anesthetized mice were positioned on a plexiglass stage and received 0.1 mL of 5% fluorescein isothiocyanate (FITC)-labeled dextran (150,000 Da; Sigma-Aldrich, Taufkirchen, Germany) in the retrobulbar venous plexus for plasma staining. The chamber window was then placed under a Zeiss Axiotech fluorescence epi-illumination microscope (Zeiss, Oberkochen, Germany) and the flap microcirculation was recorded on DVD for the offline analysis. The microscopy was performed at a constant room temperature of 22 °C. An overview of the chamber was recorded for the planimetric measurement of the perfused tissue surface area at the beginning of each microscopy. Each flap was divided into three observational zones: proximal, medial and distal to the flap base (Figure 1B). Two regions of interest (ROI) were chosen per zone, containing an arterio-venous bundle that could be identified by its morphology during each microscopy for the repeated measurements. Two adjacent capillary fields were recorded per ROI. The no-longer-perfused ROI were documented with microscopic images for as long as they could be identified. One ROI was recorded additionally within the medial transition zone between the perfused and non-perfused tissue to examine the new vessel formation.

The microscopic images were analyzed offline by means of the analysis system, CapImage (Version 8.5, Zeintl, Heidelberg, Germany). The rate of necrosis expressed in % was determined as 100-(perfused surface area/total chamber surface area ×100). Per capillary field, the functional capillary density (FCD) was measured and expressed in cm/cm^2^. Within each ROI, the microhemodynamic parameters were measured in the arterioles, capillaries and venules. The vessel diameters (D) were measured perpendicular to the vessel path in µm. Using the line shift method, the centerline red blood cell (RBC) velocity (V) was assessed [20]. The volumetric blood flow (VQ) was calculated from V and D as $\mathrm{VQ}\;=\;\pi \times {\left(\frac{D}{2}\right)}^{2}\times \frac{V}{K}\;$ with K (=1.6) representing the Baker–Wayland factor considering the parabolic velocity profile of the blood in microvessels and expressed in pL/s [21]. Angiogenesis within the transition zone was assessed by quantifying the density of newly formed microvessels, which was expressed in cm/cm^2^. The newly formed microvessels could be clearly distinguished by their irregular and entangled configuration from the straight, parallelly arranged capillaries of the panniculus carnosus muscle [22].

### 2.6. Histology and Immunohistochemistry

Tissue samples of each flap were fixed in formalin and embedded in paraffin. Three-µm-thick sections were then sliced off the processed tissue. Hematoxylin and eosin (HE) staining of individual sections was performed following standard protocol. Sections were subsequently assessed using a BX60 microscope (Olympus, Hamburg, Germany) and the imaging software cellSens Dimension 1.11 (Olympus).

For the immunofluorescent detection of microvessels, sections were stained with a monoclonal rat-anti-mouse antibody against the endothelial cell marker, CD31 (1:100; dianova GmbH, Hamburg, Germany), and with a polyclonal rabbit-anti-human antibody against the microvascular smooth-muscle cell marker, α-smooth muscle actin (α-SMA), (1:100; Abcam, Cambridge, UK) as primary antibodies. A goat-anti-rat IgG-Alexa555 antibody (1:200; Thermo Fisher Scientific, Karlsruhe, Germany) and a goat-anti-rabbit IgG-Alexa488 antibody (1:200; Thermo Fisher Scientific) served as the secondary antibodies. On each section, cell nuclei were stained with Hoechst 33,342 (2 µg/mL; Sigma-Aldrich) to merge the images exactly. The stained sections served for the assessment of the microvessel density (all CD31^+^ microvessels per high-power field (HPF)) and the fraction of CD31^+^/α-SMA^+^ microvessels (in %) in two randomized HPFs at the flap base (proximal zone) and in the medial transition zone, where the border between the vital and necrotic tissues was detectable.

For the immunohistochemical detection of the myeloperoxidase-positive (MPO^+^) neutrophilic granulocytes and cleaved caspase (Casp)-3^+^ cells undergoing apoptosis, additional sections were used. Antigens in the samples were demasked by citrate buffer and the unspecific binding sites were blocked using goat serum. Cells were stained by incubation with a polyclonal rabbit antibody against MPO (1:100; Abcam, Cambridge, UK) or a monoclonal rabbit antibody against Casp-3 (1:100; Cell signaling Technology, Danvers, MA, USA) as primary antibodies, followed by a biotinylated goat anti-rabbit IgG antibody (ready-to-use; Abcam) as a secondary antibody. The biotinylated antibody was detected with peroxidase-labeled streptavidin (ready-to-use; Abcam). The used chromogen was 3-amino-9-ethylcarbazole (Abcam). The counterstaining was performed using Mayer’s hemalum (Merck, Darmstadt, Germany). The stained cells were counted in two randomized HPFs in the proximal and medial transition zones of the flaps.

### 2.7. Statistical Analysis

Data were tested for normal distribution and equal variance. Afterwards, the two groups were analyzed for differences using the unpaired Student’s t-test (GraphPad Prism 9; GraphPad Software, San Diego, CA, USA). A Mann–Whitney rank-sum test was used in the case of non-parametric data. All values are expressed as means  ±  standard error of the mean (SEM) and the statistical significance was accepted for a value of *p*  <  0.05.

## 3. Results

### 3.1. Intravital Fluorescence Microscopy

The flap’s survival and perfusion within the dorsal skinfold chambers were analyzed using repeated intravital fluorescence microscopy. The flaps of bromelain-treated animals exhibited a significantly lower necrosis rate of ~12–15% throughout the 10-day observation period when compared to the flaps of untreated controls, which presented with a necrosis rate of ~39–44% (Figure 1C,D). This lower flap necrosis rate was associated with a significantly higher FCD in all flap zones over the entire course of the experiments (Figure 2). In the proximal and medial zones of the bromelain-treated flaps, the FCD was ~250–300 cm/cm^2^, whereas the distal zone exhibited an FCD of ~200 cm/cm^2^ (Figure 2A–D). In contrast, the FCD in the proximal and medial zones of untreated flaps was markedly reduced (~150–200 cm/cm^2^) and only measurable in the distal zone on day 1 (~85 cm/cm^2^) (Figure 2A–D).

Additionally, the diameter and centerline RBC velocities in arterioles, capillaries and venules of the flaps were measured to calculate the volumetric blood flow. In both groups, this microhemodynamic parameter slightly increased over the course of the observation period in all vessel types within the proximal and medial zones of the flaps (Table 1). In line with the data on flap necrosis, no perfused ROIs could be detected in the distal flap zone in untreated mice after day 1 (Table 1). Noteworthy, the bromelain-treated animals generally showed higher volumetric blood flows, particularly in the medial zone of the flaps (Table 1).

The new blood vessel formation was examined in the transition zone between the vital and necrotic flap tissue throughout the entire observation period. In both groups, the flap tissue displayed typical changes in the capillary architecture within this zone, starting on days 3 to 5. The capillaries dilated and exhibited irregular diameters. In addition, angiogenic sprouts grew out of the pre-existing and horizontally arranged microvessels (Figure 3A,B). Of note, the density of the neovessels was significantly higher in the flaps of bromelain-treated mice on days 7 and 10 when compared to the flaps in control animals (Figure 3C).

### 3.2. Histological and Immunohistochemical Analysis

At the end of the in vivo experiments, the flap tissue was histologically analyzed to assess the ischemia-induced morphological changes. The HE-stained sections were used for the identification of the transition zone between the proximal and the distal flap tissue. Notably, the distal zone was completely necrotic and, therefore, was excluded from further immunohistochemical analyses.

The quantification of the CD31^+^ microvessels revealed no significant differences between the groups in the proximal zone and the transition zone of the flaps, even though the observed number of CD31^+^ microvessels was slightly increased in the transition zone of bromelain-treated flaps (Figure 3D). The percentage of α-SMA^+^ arterioles out of all the CD31^+^ vessels did not show marked differences between the groups in the flaps’ proximal zone and transition zone (Figure 3E).

The identification of MPO^+^ neutrophilic granulocytes revealed, in contrast to the proximal zone, a massive neutrophilic cell invasion in the medial transition zones of the flaps in both bromelain-treated and untreated mice (Figure 4A,B). Of interest, this inflammatory reaction was markedly reduced in the bromelain-treated animals, as indicated by a significantly lower number of MPO^+^ cells/HPF when compared to controls (Figure 4B).

Moreover, apoptotic cells were identified by means of immunohistochemical Casp-3 staining. In both groups, the proximal vital zone of the flaps only contained a few apoptotic cells (Figure 4C,D). In contrast, apoptotic cell death was markedly increased in the medial transition zone. However, the bromelain treatment significantly reduced the number of Casp-3^+^ cells/HPF in this area when compared to the control treatment (Figure 4C,D).

## 4. Discussion

The development and clinical introduction of novel pharmacological compounds are often hindered by problems, such as low selectivity for target cells or high toxicity against normal cells. Therefore, phytochemicals are continuously explored as possible therapeutics because they generally exert few unwanted harmful side effects and are well-tolerated by patients [23]. Moreover, they bear the advantages of having a high availability and relative simplicity of acquisition [24].

The enzyme complex, bromelain, is one example of a phytochemical compound with several beneficial effects and few side effects, which have been used for various pathologies in the past. In our study, we demonstrated for the first time, that the anti-apoptotic, anti-inflammatory and anti-thrombotic effects of bromelain can also be observed in the critically perfused musculocutaneous flaps undergoing acute persistent ischemia. In fact, we could show that the rate of flap necrosis is significantly reduced by systemic bromelain administration, as the enzyme complex promotes nutritive tissue perfusion and suppresses inflammation and apoptotic cell death in the ischemically challenged flap tissue.

For our experiments, we used a perioperative administration protocol that would also be feasible under clinical conditions. Because flap surgery is often performed electively, bromelain administration, starting shortly before the flap elevation, could easily be implemented into the perioperative clinical routine. The effect of bromelain on the microcirculation and overall survival of critically perfused flaps undergoing necrosis, if kept untreated, was investigated in a modified murine dorsal skinfold chamber model. By combining the chamber technique with intravital fluorescence microscopy, repeated in vivo analyses of microvascular perfusion and blood vessel formation within a random-pattern musculocutaneous flap with clearly defined dimensions could be performed [19]. The results of the present study, which are consistent with the pleiotropic mode of action of most phytochemicals, suggest that in our experimental setting, bromelain targets not only one cellular mechanism, but rather modulates several pathways at once with synergistic effects, thus, adding to the overall improved outcome.

We observed that bromelain improves the nutritive perfusion of the flap tissue, as evidenced by a higher blood flow in all analyzed vessel types and flap zones in the bromelain-treated animals when compared to the untreated controls. In line with these findings, Bahde et al. [13] proved that bromelain increases the expression of endothelial nitric oxide (NO) synthase, causing vasodilatation and elevated perfusion in a model of warm hepatic ischemia in rats. In addition, bromelain has been shown to decrease the production of reactive oxygen species (ROS) that consume the vasodilator NO [25]. In the present study, the increased perfusion resulted in a maintained microcirculation with a significantly higher FCD in bromelain-treated flaps.

Furthermore, we observed a higher number of newly formed microvessels in the transition zone of bromelain-treated flaps. This effect was confirmed in our immunohistochemical analyses of the flap tissue, where we also observed an increased number of CD31^+^ microvessels in the medial flap zone, though the difference was not proven to be statistically significant. These findings are in line with several studies suggesting a pro-angiogenic activity of bromelain [26,27]. However, on the other hand, there are also studies reporting that the enzyme exerts an anti-angiogenic effect on cancer cell lines [18,28,29]. Hence, further analyses are necessary to exactly clarify the regulatory function of bromelain in the process of angiogenesis.

In addition, the microperfusion of the flap tissue may have been enhanced by the well-known anti-thrombotic and fibrinolytic effects of bromelain [30,31]. In this context, it should be considered that the lower flow rate and the inflammatory reaction within the elevated flap tissue create a pro-thrombotic environment [32,33]. The resulting microthrombi occlude capillary vessels, which further aggravates the already inadequate tissue perfusion [33]. Of interest, Metzig et al. [30] showed that incubation with bromelain completely prevents the thrombin-induced aggregation of platelets. Furthermore, in high doses, the enzyme complex downregulates both the external and internal pathways of the blood clotting system, inhibits fibrin synthesis and increases serum fibrinolytic activity [10,34]. Taken together, all these bromelain effects may contribute to the prevention of microthrombi and, thus, to the improved perfusion of the flap tissue.

The pro-apoptotic effects of bromelain have been described in the context of cancer cells, where it suppresses proliferation and induces apoptosis through the activation of the extracellular signal-regulated kinase (ERK)/AKT pathway [35]. In contrast, when used to prevent ischemic cell death, bromelain has been shown to decrease pro-apoptotic signaling pathways. For instance, Juhasz et al. [18] demonstrated an increased phosphorylation of Akt and FOXO3A after bromelain pretreatment in a murine cardiac ischemia model. This resulted in a significantly lower rate of apoptotic cardiomyocytes and a reduced infarct size, leading to an improved cardiac function after the ischemic insult. Similarly, we detected fewer apoptotic cells in the transition zone between the vital and necrotic tissues of bromelain-treated musculocutaneous flaps. This may also explain our observations that the bromelain-treated flaps contained more newly formed microvessels in their transition zones. In fact, it may be assumed that the cytoprotective effects of bromelain also increased the viability of microvessels in the transition zone, which could then serve as the origin for angiogenic vessel sprouts.

Finally, bromelain is also known to ameliorate inflammation, because it downregulates the expression of the transcription factor, nuclear factor (NF)-κB, which controls the expression of various pro-inflammatory enzymes and chemokines, such as cyclooxygenase-2, interleukin-6 or tumor necrosis factor-α [36,37]. Furthermore, the enzyme complex downregulates the expression of different immune cell surface markers that mediate the adhesion of intravascular leukocytes to the endothelium and their subsequent invasion into the surrounding tissue [12,38]. In the present study, the latter anti-inflammatory mechanism could also be observed in the critically perfused flap tissue, where the bromelain treatment resulted in a significantly lower number of invading MPO^+^ neutrophilic granulocytes into the transition zone when compared to controls.

Though our findings are promising, further research is warranted to test the efficacy of bromelain for the prevention of flap necrosis in different settings because flap survival is influenced by various factors, such as age and gender [39,40]. As capillary and artery numbers can decline in aging tissue, it may, for example, be interesting to assess whether the positive effects of bromelain on flap tissue are reproducible in aged mice [41,42]. Harder et al. [40] demonstrated that aging decreases the vascular reactivity in tissue flaps. Thus, the beneficial effect of bromelain on angiogenesis may be an interesting approach to compensate for this decreased vascular reactivity in aging patients undergoing flap surgery.

## 5. Conclusions

The present study demonstrates that perioperative-systemic bromelain administration effectively protects critically perfused flap tissue from necrosis. In fact, the enzyme is able to maintain nutritive tissue perfusion by suppressing ischemia-induced inflammation and apoptosis. When compared to other therapeutic approaches, bromelain may bear the considerable advantage that it can be easily implemented into standard clinical routines without causing severe side effects, making it a possible resource for fragile patients, such as the elderly. Therefore, future clinical trials should evaluate whether the herein observed high effectiveness of bromelain in preventing flap tissue necrosis can also be reproduced in clinical practice. If such clinical trials are successful, bromelain may be a promising therapeutic compound to reduce ischemic flap complications and related patient morbidity.

## Figures and Tables

**Figure 1 biomedicines-10-01449-f001:**
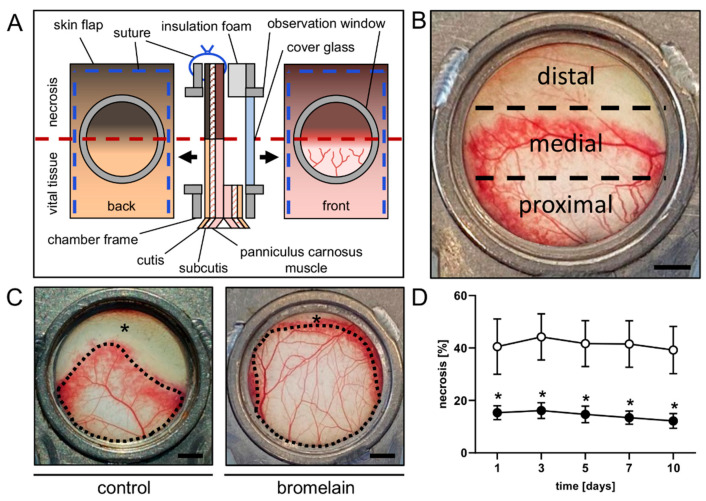
(**A**) Schematic cross-section of the musculocutaneous flap sandwiched between the symmetric titanium frames of the dorsal skinfold chamber. The observation window in one of the chamber frames provides microscopic access to the flap tissue, consisting of the panniculus carnosus muscle, subcutis and cutis. (**B**) Macroscopic image of the observation window of an untreated control mouse on day 5 after flap elevation, showing three distinct flap zones: a proximal zone consisting of well-perfused vital tissue, a medial transition zone and a distal necrotic zone. Scale bar: 2 mm. (**C**) Macroscopic images of the observation window of an untreated control mouse and a bromelain-treated mouse, displaying a significant difference in vital (borders marked by dotted line) and necrotic tissue (asterisks) on day 5 after flap elevation. Scale bar: 2 mm. (**D**) Necrosis [%] of flaps in bromelain-treated mice (black circles, *n* = 7) and untreated controls (white circles, *n* = 7) on days 1, 3, 5, 7 and 10 after flap elevation, as assessed by intravital fluorescence microscopy and computer-assisted image analysis. Means  ±  SEM. * *p*  <  0.05 vs. control.

**Figure 2 biomedicines-10-01449-f002:**
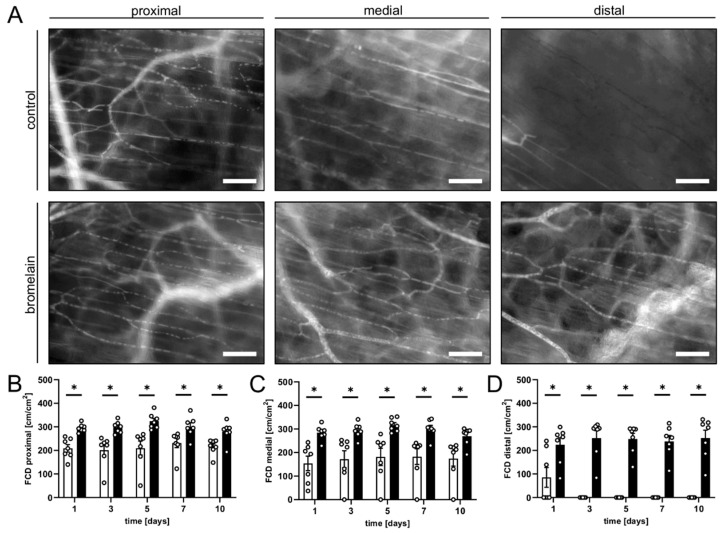
(**A**) Intravital fluorescent microscopic images of the proximal, medial and distal zones of flaps in an untreated control mouse and a bromelain-treated mouse on day 5 after flap elevation. Scale bar: 50 μm. (**B**–**D**) FCD [cm/cm^2^] in the proximal (**B**), medial (**C**) and distal zones (**D**) of flaps in bromelain-treated mice (black bars, *n* = 7) and untreated controls (white bars, *n* = 7) on days 1, 3, 5, 7 and 10 after flap elevation, as assessed by intravital fluorescence microscopy and computer-assisted image analysis. Means  ±  SEM. * *p*  <  0.05 vs. control.

**Figure 3 biomedicines-10-01449-f003:**
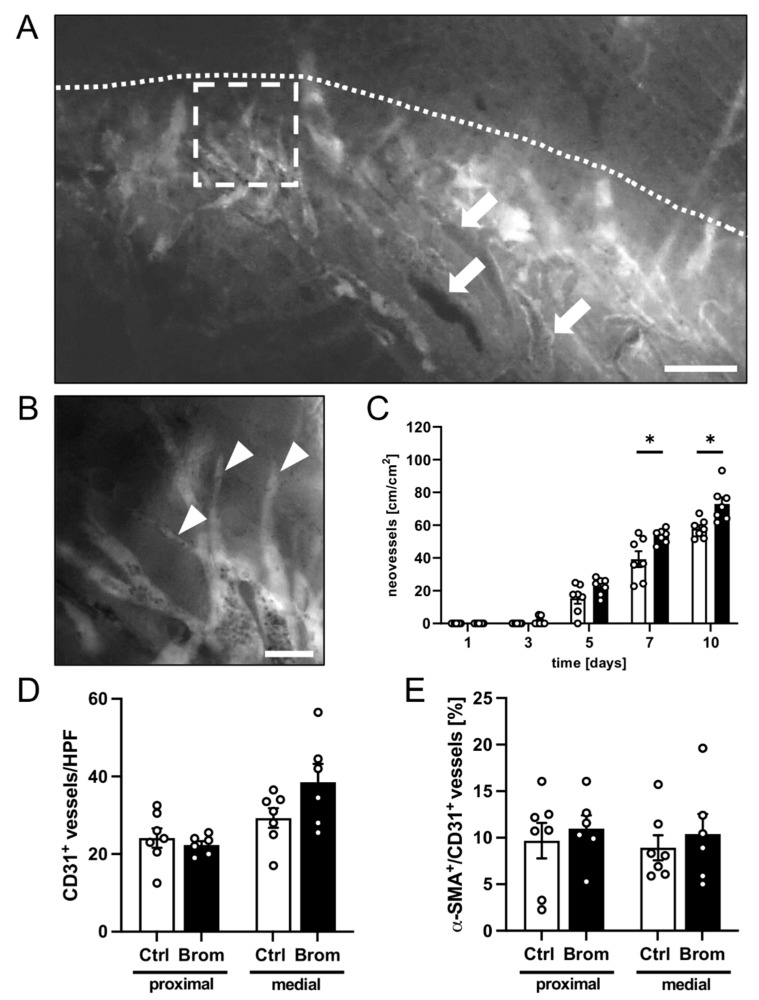
(**A**) Intravital fluorescent microscopic images of the medial zone of a flap in a bromelain-treated mouse on day 10. Note the transition zone with dilated vessels (arrows) adjacent to the necrotic tissue (border marked by a dotted line). Scale bar: 200 µm. (**B**) Higher magnification of insert in image A (borders marked by a broken line), which displays angiogenic sprouts (arrowheads) developing from pre-existing perfused vessels. Scale bar: 50 μm. (**C**) Neovessels [cm/cm^2^] in the medial transition zones of flaps in bromelain-treated mice (black bars, *n* = 7) and untreated controls (white bars, *n* = 7) on days 1, 3, 5, 7 and 10 after flap elevation, as assessed by intravital fluorescence microscopy and computer-assisted image analysis. (**D**) CD31^+^ vessels/HPF in the proximal and medial zones of flaps in bromelain-treated mice (Brom; black bars, *n* = 6) and untreated controls (Ctrl; white bars, *n* = 7) on day 10 after flap elevation, as assessed by immunohistochemistry. (**E**) Percentage of α-SMA^+^ vessels out of all CD31^+^ vessels per HPF in the proximal and medial zones of flaps in bromelain-treated mice (Brom; black bars, *n* = 6) and untreated controls (Ctrl; white bars, *n* = 7) on day 10 after flap elevation, as assessed by immunohistochemistry. Means  ±  SEM. * *p*  <  0.05 vs. control.

**Figure 4 biomedicines-10-01449-f004:**
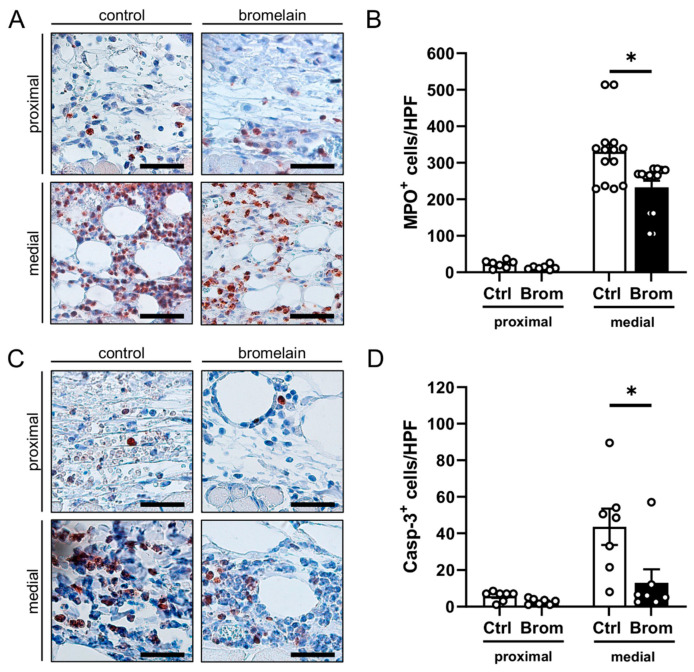
(**A**) Microscopic images of immunohistochemical sections of the proximal and medial zones in flaps of an untreated control mouse and a bromelain-treated mouse on day 10 after flap elevation. The sections were stained with an antibody against the neutrophilic granulocyte marker, MPO. Scale bar: 50 µm. (**B**) MPO^+^ cells/HPF in the proximal and medial zones of flaps in bromelain-treated mice (Brom; black bars, *n* = 7) and untreated controls (Ctrl; white bars, *n* = 7) on day 10 after flap elevation, as assessed by immunohistochemistry. (**C**) Microscopic images of immunohistochemical sections of the proximal and medial zones of flaps in an untreated control mouse and a bromelain-treated mouse on day 10 after flap elevation. The sections were stained with an antibody against the apoptotic marker, Casp-3. Scale bar: 50 µm. (**D**) Casp-3^+^ cells/HPF in the proximal and medial zones of flaps in bromelain-treated mice (Brom; black bars, *n* = 7) and untreated controls (Ctrl; white bars, *n* = 7) on day 10 after flap elevation, as assessed by immunohistochemistry. Means  ±  SEM. * *p*  <  0.05 vs. control.

**Table 1 biomedicines-10-01449-t001:** Volumetric blood flow [pL/s] of arterioles, capillaries and venules in the proximal, medial and distal zones of flaps in untreated control mice (Ctrl; *n* = 7) and bromelain-treated mice (Brom; *n* = 7) on days 1, 3, 5, 7 and 10 after flap elevation, as assessed by intravital fluorescence microscopy and computer-assisted image analysis. Means  ±  SEM. * *p*  <  0.05 vs. control.

Volumetric Blood Flow [pL/s]		d1	d3	d5	d7	d10
**Arterioles**						
**proximal**	**Ctrl**	458 ± 93	598 ± 158	814 ± 152	1145 ± 230	1272 ± 160
	**Brom**	790 ± 86 *	807 ± 127	960 ± 146	1459 ± 196	1634 ± 160
**medial**	**Ctrl**	437 ± 96	603 ± 117	733 ± 90	1013 ± 152	1085 ± 159
	**Brom**	687 ± 107	1036 ± 91 *	1302 ± 165 *	1574 ± 120 *	1550 ± 188
**distal**	**Ctrl**	485 ± 149	-	-	-	-
	**Brom**	538 ± 116	962 ± 237	1141 ± 276	1203 ± 261	1540 ± 248
**Capillaries**						
**proximal**	**Ctrl**	3 ± 0	3 ± 0	4 ± 1	5 ± 1	6 ± 1
	**Brom**	4 ± 0	4 ± 0 *	4 ± 1	6 ± 1	7 ± 1
**medial**	**Ctrl**	3 ± 0	3 ± 0	4 ± 1	5 ± 1	6 ± 1
	**Brom**	4 ± 0	4 ± 0 *	4 ± 1	6 ± 1	7 ± 1
**distal**	**Ctrl**	2 ± 1	-	-	-	-
	**Brom**	2 ± 0	4 ± 0	5 ± 1	6 ± 1	6 ± 1
**Venules**						
**proximal**	**Ctrl**	383 ± 47	502 ± 89	667 ± 100	776 ± 137	1233 ± 517
	**Brom**	773 ± 87 *	1167 ± 131 *	1570 ± 410 *	1894 ± 536 *	1876 ± 397
**medial**	**Ctrl**	333 ± 47	452 ± 74	691 ± 140	897 ± 200	779 ± 266
	**Brom**	833 ± 158 *	1265 ± 146 *	1766 ± 389 *	2009 ± 362 *	1650 ± 433
**distal**	**Ctrl**	146 ± 41	-	-	-	-
	**Brom**	472 ± 171	1012 ± 308	1387 ± 326	1268 ± 301	1859 ± 337

## Data Availability

Data is contained within the article.

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
