# Peer review of "Bromelain Protects Critically Perfused Musculocutaneous Flap Tissue from Necrosis"

_biomedicines, 2022, doi:10.3390/biomedicines10061449_

Round 1
Reviewer 1 Report
Authors attempt to demonstrate that bromelain prevents flap necrosis by maintaining nutritive tissue perfusion and suppressing ischemia-induced inflammation and apoptosis.
The study has some novelty. However, it would be important to strenghten the findings by deepening some aspects:
for instance, they could provide additional info regarding the quantification of capillary and arterioles (CD31 and smooth muscle active positive) on the histogical samples).
Any idea of other mechanisms involved? such as VEGF?
any dose-dependent effects?
the effect is it only related on the decrease of apopotosis or a parallel enhancement of angiogenesis also occurs?
Author Response
We appreciate the fair and constructive comments of the reviewer. In the following, please find our detailed reply.
We have revised our manuscript according to the suggestions of reviewer 1 and have added further analysis regarding the quantification of capillaries and arterioles in the form of additional immunohistochemical stainings for CD31 and α smooth muscle actin (α-SMA). We did observe a higher amount of CD31+ microvessels in the medial transition zone of bromelain-treated flaps, however, the difference was not statistically significant. The share of α-SMA+arterioles out of all CD31+ microvessels did not show marked differences in the proximal and medial flap zones (see revised figure 3; page 5, lines 182-194; page 8, lines 268-273; page 9, lines 281-286; page 11, lines 349-357; page 13, lines 480-490; marked in yellow). Though the investigation of the effect of bromelain on angiogenesis, including VEGF as a possible pathway to mediate such effects, was not the primary focus of the present study, we concur with the reviewer that it would be an interesting topic for further research and we will try to include it into our next study.
Reviewer 2 Report
The authors present very exciting and important findings on bromelain - which prevents flap necrosis by maintaining tissue perfusion and suppressing ischemia-induced inflammation and apoptosis. The experiments are performed well, and the data is robust.
Age and sex play an important role in ischemia and ischemic stroke pathology. Also, ageing is known to impact the tissue microenvironment, perfusion and vasculature. These experiments in the manuscript are performed on young male mice. What is the outcome if bromelain aged mice are used? Ageing is associated with the loss of pericytes and blood vessels. Does bromelain treatment have a different effect in young versus aged mice with ischemia? Also, factors like ageing and sex should be included and discussed and the relevant literature such as PMID: 33215738, and PMID: 33536212 should be cited.
Author Response
We appreciate the fair and constructive comments of the reviewer. In the following, please find our detailed reply.
We have revised our manuscript according to the suggestions of reviewer 2 and have added more information on the influence of gender and age on the microperfusion of flaps into the discussion section to round out our manuscript. We have also included the suggested literature in our revised manuscript (see page 11, lines 393-401; page 14, lines 514-524; marked in yellow). The additional repetition of animal experience with aged mice was not possible due to ethical and legal considerations, as an additional application with the local government would be necessary to perform such experiments. Nonetheless, we feel that the suggestions of the reviewer are highly useful for follow-up studies.
Round 2
Reviewer 1 Report
The reviewer has been satisfied
Reviewer 2 Report
The authors have addressed my comments and I have no further comments!